# Role of Intracellular Pulmonary Pathogens during SARS-CoV-2 Infection in the First Pandemic Wave of COVID-19: Clinical and Prognostic Significance in a Case Series of 1200 Patients

**DOI:** 10.3390/microorganisms10081636

**Published:** 2022-08-12

**Authors:** Matteo Guarino, Benedetta Perna, Francesca Cuoghi, Michele Domenico Spampinato, Alice Eleonora Cesaro, Francesca Manza, Adriana Pretula, Anastasio Grilli, Martina Maritati, Giacomo Caio, Aldo Carnevale, Maria Elena Flacco, Roberto De Giorgio, Carlo Contini

**Affiliations:** 1Department of Translational Medicine, St. Anna University Hospital of Ferrara, University of Ferrara, 44124 Ferrara, Italy; 2Infectious and Dermatology Diseases, St. Anna University Hospital of Ferrara, University of Ferrara, 44124 Ferrara, Italy; 3Department of Environmental and Preventive Sciences, St. Anna University Hospital of Ferrara, University of Ferrara, 44124 Ferrara, Italy

**Keywords:** COVID-19, in-hospital mortality, co-infections, *Chlamydia pneumoniae*, *Mycoplasma pneumoniae*

## Abstract

*Background*: Since 2019, the severe acute respiratory syndrome coronavirus 2 (SARS-CoV-2) pandemic (COVID-19) has caused millions of deaths worldwide and is the second most serious pandemic after the Spanish flu. Despite SARS-CoV-2 infection having a dominant effect on morbidity and life-threatening outcomes, the role of bacterial co-infection in patients with COVID-19 is poorly understood. The present study aimed to verify the existence of bacterial co-infections and their possible role as cofactors worsening COVID-19-related clinical manifestations. *Methods*: All patients with suspected SARS-CoV-infection, hospitalised in COVID-19 wards at the Sant’Anna University Hospital of Ferrara, were retrospectively included in this single-centre study and their specific bacterial serologies were assessed. Univariate and logistic regression analyses were performed. *Results*: A total of 1204 individual records were retrieved. Among them, 959 were excluded because of a negative nasopharyngeal swab or missing data; of the eligible 245 patients, 51 were co-infected. Compared to patients with SARS-CoV-2 infection alone, those with *Chlamydia pneumoniae* or *Mycoplasma pneumoniae* co-infections had worse respiratory/radiological features and more intensive care unit admissions. However, the co-infection did not result in a higher mortality rate. *Conclusions*: The present study, comparing clinical, laboratory and radiological findings between patients with COVID-19 vs. those with co-infections (*C. pneumoniae* or *M. pneumoniae*) showed that, on admission, these features were worse in co-infected patients, although the mortality rate did not differ between the two groups.

## 1. Introduction

Coronavirus Disease 2019 (COVID-19) is a life-threatening disease caused by the SARS-CoV-2 virus, and it was firstly reported in China and then spread worldwide [1]. Since the beginning of the COVID-19 pandemic, millions of patients have died because of complicated SARS-CoV-2 pneumonia [1,2,3]. The clinical features of COVID-19 range from an asymptomatic condition to severe/fatal lung injury and multi-organ failure. Common complications include acute respiratory distress syndrome (ARDS), acute kidney and liver dysfunctions, delirium/encephalopathy, thrombosis, and cardiac damage (e.g., cardiomyopathy, arrhythmias, and sudden cardiac death) [1,2]. The severity of this condition suggested that pulmonary co-infections (e.g., *Chlamydia pneumoniae*, *Mycoplasma pneumoniae*, pneumococcus, or other agents, e.g., viruses) might have a role in worsening clinical manifestations of SARS-CoV-2 pneumonia [4]. In particular, *C. pneumoniae* (CP) and *M. pneumoniae* (MP) are responsible for atypical community acquired pneumonia and extra-pulmonary manifestations in adults and children [5,6,7]. However, so far, few studies have addressed the role of bacterial co-infections as contributory factors worsening the respiratory distress associated with COVID-19 and the subsequent outcomes [4]. Since the diagnosis of co-infections with CP or MP based on clinical presentation and radiological findings is often challenging, serological investigation is mandatory for diagnosis [8,9].

The present study has been undertaken to evaluate two aspects: (*i*) the possible co-infection of MP and CP in COVID-19 by the serological assessment of specific antibodies; and (*ii*) the possible role of both MP and CP as cofactors worsening COVID-19 clinical manifestations and outcomes.

## 2. Materials and Methods

Between March and October 2020, the anonymous medical records of patients with suspected SARS-CoV-2 infection admitted to the Intensive Care Unit (ICU), Pneumology, Infectious Diseases, and Internal Medicine wards of the Sant’Anna University Hospital of Ferrara were retrieved. SARS-CoV-2 was detected by nasopharyngeal swab, while the assessment of chlamydial and mycoplasma co-infection was performed during the first 48/72 h after admission using a serological assay for the determination of IgG and IgM class antibodies against *Chlamydia pneumoniae* and *Mycoplasma pneumoniae* (LIAISON^®^
*Mycoplasma pneumoniae* IgG and IgM, DiaSorin, Ireland and VIRCLIA^®^ IgG, IgM and IgA Chlamydophila pneumoniae, Vircell, Spain, respectively) according to the manufacturer’s instructions. Patients who tested negative for SARS-CoV-2 or whose tests were not recoverably detected were excluded from the study. The enrolled cohort was divided into three subgroups: group A (SARS-CoV-2 infection only); group B (SARS-CoV-2 and MP); and group C (SARS-CoV-2 and CP).

CT scans performed at admission were analysed by a radiologist with experience in chest imaging, blinded to clinical data, according to the COVID-19 pneumonia imaging classification system proposed by the Radiological Society of North America (RSNA) (grading 0 to 3) [10] and the COVID-19 Reporting and Data System (CO-RADS: category grading 0 to 5) [11]. The total disease extent was scored by visual assessment (from 0 to 4: 1 = 0–24% of the parenchyma; 2 = 25–49%; 3 = 50–74%; and 4 = 75–100%); elementary alterations were analysed regarding: consolidation, ground glass opacity, and crazy paving (extent assessed individually as above); ‘tree in bud’ opacities (sign of distal airway inflammation) were graded from 0 to 6 (for the number of lobes, considering lingula separately); pleural effusion was graded from 0 to 2 (1 = unilateral; 2 = bilateral); pericardial effusion rating (0/1); and lymphadenomegaly (0/1). Other parameters considered were: (i) vital signs at the emergency room (blood pressure, HR, SpO_2_, FiO_2_, RR, and temperature); (ii) laboratory tests (white blood cells, neutrophils, lymphocytes, monocytes, hemoglobin, platelets, fibrinogen, aPTT, creatinine, urea, D-dimer, total and direct bilirubin, Na, K, CK, LDH, CRP, and procalcitonin); and (iii) arterial blood gas (ABG) (pH, pCO_2_, pO_2_, HCO_3_, and lactates).

The outcomes analysed included: in-hospital mortality (IHM), 7- and 30-day mortality, length of stay (LOS), and hospitalization in ICU (Table 1 and Table 2). Furthermore, a comparative analysis among involved groups was performed considering the Charlson Comorbidity Index (CCI), age, and sex.

This was a retrospective observational study conducted only through a careful analysis of medical charts/records without direct involvement with or identification of the patients. Thus, in agreement with our local ethics committee (Comitato Etico di Area Vasta Emilia Centro (CE-AVEC) of Azienda Ospedaliero Universitaria di Ferrara) a formal approval for this study was deemed unnecessary and, thereby, a project identification code was not assigned.

### Statistical Analysis

Standard univariate analyses were used to detect differences in each clinical, laboratory, and radiological parameter among the involved groups. The Chi-squared and Fisher’s exact tests were performed for categorical variables; the t-test was performed for normally distributed variables and Kruskal–Wallis test for non-normally distributed continuous variables; and the distribution was evaluated with the Shapiro–Wilk test. Multivariate analyses (Table 3) were performed to evaluate the potential predictors of each outcome in group B (subjects co-infected with SARS-CoV-2 and MP) and C (subjects co-infected with SARS-CoV-2 and CP) vs. group A (SARS-CoV-2 infections only), respectively. Due to the limited number of co-infected patients, the analyses were repeated with the same approach described above considering all coinfected subjects together vs. SARS-CoV-2 infected subjects (Table 4). Statistical significance was defined as a *p*-value of <0.05. All analyses were carried out using Stata, version 13.1 (Stata Corp., College Station, TX, USA, 2014).

## 3. Results

A total of 1204 records of patients with suspected SARS-CoV-2 infection were retrieved. Despite the clinical and radiological signs suggestive of COVID-19 pneumonia, only 245 (20.3%) tested positive at nasopharyngeal swab and were included in this analysis. The eligible cohort (245 patients) was divided into three groups: (A) patients testing positive for SARS-CoV-2 and with related COVID-19 disease alone (*n* = 194); (B) patients with COVID-19 and MP (*n* = 32); and (C) patients with COVID-19 and CP (*n* = 19). The analysis did not show any statistical differences in terms of age and sex among the three analysed groups (Table 1).

The comorbidities assessment (CCI) showed a score of ≥ 2 in 33.5% of patients in group A vs. 12.5% in group B and 5.3% in group C. The most common comorbidities in group A were acute myocardial ischemia (AMI), chronic heart failure (CHF), vascular cerebropathy and dementia, chronic pulmonary disease, rheumatic disease, diabetes, renal failure, and cancer. The main comorbidities in group B were AMI, chronic pulmonary disease, vascular cerebropathy and dementia, diabetes, and renal failure; and in group C, a higher number of vascular cerebropathies, dementia, and tumors was found. The prevalence of risk factors such as obesity and hypertension was not significantly different in the three groups.

Among the lab tests, fibrinogen was statistically higher in group B (590 mg/dL) and C (632 mg/dL) than in group A (542 mg/dL) (*p* < 0.05). Arterial pO_2_ (mmHg), measured by ABG analysis, showed values of 76.8 (A), 66.6 (B), and 61.2 (C), respectively, with statistical significance in the comparison between group A and C. The other findings are summarised in Table 1.

A CO-RADS score of 5 and a Ground Glass Opacity score of ≥ 3 were reported more frequently in group C than in group A (*p* < 0.05).

Although no statistically significance has been reported, our study showed a higher mortality (i.e., IHM and 7- and 30-day mortality) and length of stay (LOS) in group A vs. B and C (Table 1). ICU admission was higher in patients with co-infections, particularly in group C.

In the multivariate analysis, age had a negative impact on 30-day mortality (Table 2 a/b). Furthermore, in MP co-infection, male sex (OR 3.96, 95% CI, 1.37–11.5, *p* = 0.011) was a negative predictive factor for ICU admission.

Comparisons between group A and groups B and C (*n* = 51) are shown in Table 3. Arterial pO_2_ was significantly lower in co-infected patients (64.5 vs. 76.8 mmHg, *p* = 0.002).

The multivariate analysis showed that age is a negative predictive factor for IHM and 7- and 30-day mortality. The other findings are shown in Table 4.

## 4. Discussion

The present study showed that co-infection induced by MP and CP can lead to a worsening of the clinical presentation at the ED in COVID-19 patients, although no differences in IHM were observed. Several studies showed that respiratory viral infections can overlap with bacterial ones, worsening the clinical conditions and increasing morbidity and mortality [12]. At the beginning of the pandemic, the coexistence of bacterial intracellular lung pathogens seemed rare, even if this association could occur, as previously reported [4,8]. Therefore, the isolation of SARS-CoV-2 is not sufficient to exclude the presence of other respiratory pathogens that could already be involved and potentially responsible for the worsening of SARS-CoV-2-related interstitial pneumonia due to their overlapping on the lung damage triggered by the virus [13,14].

Zhou et al. reported that 50% of patients with COVID-19 (that died from COVID-19) had bacterial co-infections [8,12]. The presence of co-infections is supposed to affect the clinical manifestations and the prognosis of COVID-19 related pneumonia. Different pathogens have been identified, including bacteria (e.g., Streptococcus pneumoniae, Staphylococcus aureus, Klebsiella pneumoniae, *Mycoplasma pneumoniae*, *Chlamydia pneumoniae*, Legionella pneumophila, and Acinetobacter baumannii), mycetes (e.g., Candida albicans and Aspergillus flavus), and viruses (e.g., influenza, coronavirus, rhinovirus/enterovirus, parainfluenza, metapneumovirus, influenza B, human immunodeficiency virus (HIV), and HIV-opportunistic related infections such as *P. jiroveci*) [4]. Among these co-infections, we mainly focused on intracellular bacteria (i.e., MP and CP) which are prevalent in community settings and responsible for interstitial pneumonia. In our study, we are confident that these were co-infections and not secondary infections because the anti-MP and -CP serologic tests performed within 48 h of the patients’ admissions showed IgM titers above the cut-off values. These findings demonstrate that the infections were not contracted in the nosocomial setting. One can assume that patients already had the bacterial infection at the time of hospital admission since IgM generally appeared about 10 days after infection. Alternately, one may speculate that a bacterial infection contracted before SARS-CoV-2 exposure caused the pulmonary impairment responsible for a more severe COVID-19 presentation.

In the enrolled cohort, 20.8% of patients tested positive for anti-MP and anti-CP IgMs. The detection of co-infections can be challenging due to the lack of accurate diagnostic tests, peculiar radiological features, and suggestive laboratory alterations. Nowadays, this diagnosis is based on serological assessment with specific chemiluminescence and immuno-enzymatic tests that represent the diagnostic gold standard for these infections [15,16].

Some viral pandemic pneumonias may present with similar clinical and radiological features that are difficult to distinguish from those of bacterial origin. (1) During the first wave (and, to some extent, the second wave) of pandemic, suspected COVID-19 patients underwent lung CT scans, routine blood tests, and MP/CP serology. The urinary antigens for Streptococcus pneumoniae and Legionella pneumophila always resulted negative. Other techniques were limited because of the massive inflow of patients. (2) Pneumocystis jiroveci infection was ruled out in the most severe cases on the basis of specific antigenic tests in respiratory secretions [17]. Furthermore, pneumocystis-related pneumonia generally occurs in severely immunocompromised patients with very low CD4+ levels. However, this event did not occur in our cohort. The comparison among the involved groups showed that respiratory (in terms of pO_2_) and radiological impairment were more severe in co-infected patients. These findings suggest that the presence of intracellular pathogens can worsen the patients’ clinical conditions even though the CCIs were less compromised in groups B and C than they were in A. Furthermore, despite a higher ICU hospitalization rate among patients with co-infections, the overall mortality in these groups was inferior to those infected with COVID-19 alone. This result may be explained by a lower rate of comorbidities and the prompt administration of specific antibiotics (in particular, azithromycin, which was used prophylactically during the first wave of SARS-CoV-2 infections) and anticoagulants.

The laboratory tests showed that co-infections caused an increase in D-dimer and fibrinogen values, causing a higher risk of thrombosis. MP and CP are bacteria known to evoke hypercoagulability [18,19]. Furthermore, co-infections increase the risk of thrombosis compared to patients with COVID-19 infection alone. The D-dimer assessment is strongly recommended in routine anticoagulant prophylaxis [16,17,18,19,20].

Comparing clinical, laboratory, and radiological findings in both groups B and C, chlamydia co-infected patients had worse clinical conditions, probably due to a higher virulence and a stronger hyper-coagulating activity. In fact, according to some authors, chlamydia appears to be a potent inducer of a pro-coagulative state of the physiologically inert vasculature, with a potential increased risk for acute thrombotic events in people subjected to chlamydial infection [21].

If the results were confirmed on a larger sample, the early identification of any pathogen superimposing to SARS-CoV-2 infection would be particularly significant to improve patients’ outcomes.

This study has three main limitations: (*i*) Not all hospitalised patients with SARS-CoV-2 infection were tested for MP and CP. (*ii*) Since there was a limited number of cases, it is very difficult to determine the role of these pathogens in COVID-19 co-infected patients. Thus, the serological analysis was based on specific anti-MP or anti-CP IgM antibodies with the risk of false positive results, which should always be taken into account when deciding to rule out SARS-CoV-2 infection; moreover, a further limitation was the lack of available paired samples to confirm previous serological results for the diagnosis of atypical pathogens. (*iii*) The diagnosis of co-infections was based on serological response only as the molecular analysis of respiratory samples for the detection of the *M. pneumoniae* or *C. pneumoniae* genome was not possible, and thus these samples were not available [22].

## 5. Conclusions

The study supports the theory that patients with COVID-19 may have had co-infections caused by intracellular agents, i.e., MP and CP, during the first wave of pandemic in 2020. Our results also indicated the bacterial co-infection of COVID-19 in about 21% of patients (51 out of 245) using anti-CP and -MP serology, along with a number of other relevant parameters emerging from radiological imaging and laboratory tests (e.g., D-dimer and fibrinogen elevation vs. COVID-19 infection only). The groups of patients with co-infection had lower CCIs than patients with COVID-19 infection only. Furthermore, being of the female gender was confirmed be a protective factor against COVID-19. The most plausible hypothesis for the higher mortality rate in group A vs. B and C is that bacterial co-infections in COVID-19 patients were promptly treated with effective antibiotics (e.g., macrolides). Although later confirmed to be ineffective [23], this treatment initially seemed [24] to be active against SARS-CoV-2, in addition to timely anticoagulant administration needed because of the hyper-coagulant effects evoked by the two pathogens.

Clearly, further prospective studies on larger sample sizes are awaited to expand our results and provide thorough explanations of the different mortality rates observed in the present study.

## Figures and Tables

**Table 1 microorganisms-10-01636-t001:** Selected characteristics and outcomes of the sample according to infection status: (a) SARS-CoV-2 infection only; (b) SARS-CoV-2 and MP co-infection; and (c) SARS-CoV-2 and CP co-infection.

	SARS-CoV-2 Infection Only	MP Co-Infection	CP Co-Infection	*p* **
Variables	(*n* = 194)	(*n* = 32)	(*n* = 19)	
Male gender, %	51.0	46.9	63.2	
Mean age (SD)	66.7 (18.1)	65.1 (19.2)	71.6 (12.6)	
Median Charlson index at admission (IQR)	1.0 (3.0)	1.0 (2.0)	0.0 (2.0)	
- % patients with score ≥ 2	33.5	12.5	5.3	^A,B^
- % patients with score ≥ 5	8.3	6.3	0.0	
Obesity, %	11.9	12.5	10.5	
Hypertension, %	49.5	34.4	57.9	
Steatosis, %	3.4	3.2	0.0	
Viral hepatitis, %	1.6	6.3	0.0	
*Laboratory parameters, mean (SD) **				
WBC/24 hrs after admission, ×10^3^/µL	7.77 (8.10)	8.02 (5.10)	6.89 (3.15)	
Haemoglobin, g/dL	12.9 (2.3)	12.5 (2.2)	12.7 (2.3)	
PLT, ×10^3^/µL	199 (87)	225 (103)	238 (57)	
Neutrophils, ×10^3^/µL	5.47 (3.64)	6.76 (5.38)	4.48 (2.15)	
Lymphocytes, ×10^3^/µL	1.36 (3.11)	1.23(0.68)	1.16 (0.61)	
Monocytes, ×10^3^/µL	0.90 (4.91)	0.52 (0.33)	0.55 (0.39)	
PT, mL/min	1.12 (0.39)	1.09 (0.11)	1.10 (0.19)	
Fibrinogen, mg/dL	542 (157)	590 (183)	632 (143)	^B^
APTT, mL/min	1.01 (0.18)	1.02 (0.11)	1.03 (0.18)	
Creatinine, mg/dL	1.27 (1.18)	0.99 (0.26)	1.02 (0.46)	
Urea, mg/dL	56.9 (47.2)	49.5 (54.0)	57.5 (31.0)	
Bilirubin tot, mg/dL	0.67 (0.33)	0.81 (0.70)	0.57 (0.11)	
Bilirubin dir, mg/dL	0.19 (0.13)	0.25 (0.17)	0.16 (0.06)	
Na, mmol/L	137 (5.12)	137 (5.22)	138 (5.58)	
K, mmol/L	3.93 (0.61)	3.82 (0.45)	4.22 (0.32)	
D-dimer, mg/L	1.96 (3.98)	2.27 (3.57)	4.21 (8.55)	
Lactate, mmol/L	1.21 (0.71)	1.03 (0.41)	0.83 (0.31)	
LDH, U/L	326 (146)	366 (142)	349 (132)	
CRP, mg/dL	11.3 (44.5)	8.07 (7.54)	9.38 (7.56)	
CPK, U/L	235 (617)	166 (189)	132 (128)	
Procalcitonin, ng/mL	1.75 (5.47)	1.21 (2.28)	0.36 (0.60)	
pO_2_, mmHg	76.8 (27.7)	66.6 (13.3)	61.2 (15.0)	^B^
pCO_2_, mmHg	33.5 (6.11)	31.9 (3.2)	32.0 (6.4)	
*CT parameters*				
Pneumonia, %	29.9	25.8	36.8	
Median RSNA category (IQR)	3.0 (1.0)	3.0 (0.0)	3.0 (0.0)	
- % patients with score = 3	64.4	76.9	88.2	
Median CO-RADS category (IQR)	5.0 (2.0)	5.0 (0.0)	5.0 (0.0)	
- % patients with score = 5	62.0	76.9	88.2	^B^
Median extent (IQR)	2.0 (1.0)	1.0 (2.0)	2.0 (1.0)	
- % patients with score ≥ 3	31.3	42.3	47.0	
Median consolidation (IQR)	1.0 (1.0)	1.0 (1.0)	1.0 (1.0)	
- % patients with score ≥ 3	1.8	3.9	5.9	
Median ground-glass opacity (IQR)	1.0 (1.0)	1.0 (1.0)	1.0 (2.0)	
- % patients with score ≥ 3	1.8	9.3	11.9	^B^
Median crazy paving (IQR)	0.0 (1.0)	1.0 (1.0)	1.0 (1.0)	
- % patients with score ≥ 3	6.7	11.5	5.9	
Tree in bud ≥ 2, %	1.8	3.9	0.0	
Pleural effusion (mono/bilateral), %	18.4	3.9	5.9	
Pericardial effusion, %	1.8	0.0	0.0	
Lymph node enlargement, %	17.2	15.4	29.4	
*Outcomes*				
1-week mortality, %	7.3	3.1	5.3	
30-day mortality, %	19.2	12.5	5.3	
Intra-hospital mortality, %	25.9	15.6	21.0	
Mean LOS (SD)	16.3 (18.5)	14.0 (16.2)	15.1 (10.4)	
ICU admission, %	10.8	12.9	15.8	

**Legend**. SD: standard deviation; IQR: interquartile range; WBC: white blood cells; PLT: platelets; PT: prothrombin time; APTT: activated partial thromboplastin time; CPK: creatine phosphokinase; LDH: lactate dehydrogenase; CRP: C-reactive protein; CT: computed tomography; RSNA: Radiological Society of North America; CO-RADS: COVID-19 Reporting and Data System score; ICU: intensive care unit; LOS: length of stay. * All values are means (SD) when not otherwise reported. ^A^: *p* < 0.05 for comparisons between patients with SARS-CoV-2 infections only and patients with mycoplasma co-infections; ^B^: *p* < 0.05 for comparisons between patients with SARS-CoV-2 infections only and patients with chlamydia co-infections. ** Chi-square test for categorical values; *t*-test and Kruskal–Wallis test for normally distributed and non-normally distributed continuous variables, respectively.

**Table 2 microorganisms-10-01636-t002:** Univariate and multivariate analysis evaluating the potential predictors of each outcome in subjects with SARS-CoV-2 and (a) MP co-infection or (b) CP co-infection vs. subjects with SARS-CoV-2 infection only.

	**MP Co-Infection (*n* = 32)**			
* **Outcomes** *	**Crude OR** **(95% CI)**	* **p** *	**Adjusted OR** **(95% CI)**	* **p** *
*A. 1-week mortality*				
- Co-infection (vs. SARS-CoV-2 infection only)	0.41 (0.05–3.25)	0.4	0.92 (0.10–8.22)	0.9
- Age, 1-year increase	--	--	1.04 (0.99–1.09)	0.08
- Male gender	--	--	0.48 (0.14–1.65)	0.2
- Charlson index > 2 (vs. ≤ 2)	--	--	2.39 (0.68–8.34)	0.2
*B. 30-day mortality*				
- Co-infection (vs. SARS-CoV-2 infection only)	0.60 (0.20–1.82)	0.4	0.92 (0.23–3.65)	0.9
- Age, 1-year increase	--	--	1.05 (1.02–1.08)	0.001
- Male gender	--	--	1.16 (0.54–2.51)	0.7
- Charlson index > 2 (vs. ≤ 2)	--	--	1.86 (0.83–4.17)	0.13
*C. Intra-hospital mortality*				
- Co-infection (vs. SARS-CoV-2 infection only)	0.53 (0.19–1.44)	0.2	0.93 (0.26–3.34)	0.9
- Age, 1-year increase	--	--	1.05 (1.03–1.08)	<0.001
- Male gender	--	--	1.01 (0.49–2.05)	0.9
- Charlson index > 2 (vs. ≤ 2)	--	--	2.29 (1.09–4.83)	0.03
*D. ICU admission*				
- Co-infection (vs. SARS-CoV-2 infection only)	1.22 (0.39–3.83)	0.7	0.87 (0.18–4.20)	0.9
- Age, 1-year increase	--	--	0.02(0.99–1.05)	0.9
- Male gender	--	--	3.96 (1.37–11.5)	0.011
- Charlson index > 2 (vs. ≤ 2)	--	--	0.24 (0.06–0.93)	0.04
	**CP Co-Infection (*n* = 19)**			
* **Outcomes** *	**Crude OR** **(95% CI)**	** *p* **	**Adjusted OR** **(95% CI)**	** *p* **
*A. 1-week mortality*				
- Co-infection (vs. SARS-CoV-2 infection only)	0.71 (0.09–5.72)	0.7	1.16 (0.13–10.4)	0.9
- Age, 1-year increase	--	--	1.06 (1.02–1.11)	0.007
- Male gender	--	--	--	--
- Charlson index > 2 (vs. ≤2)	--	--	--	--
*B. 30-days mortality*				
- Co-infection vs. SARS-CoV-2 infection only	0.23 (0.03–1.81)	0.2	0.25 (0.03–2.09)	0.2
- Age, 1-year increase	--	--	1.06 (1.03–1.09)	<0.001
- Male gender	--	--	--	--
- Charlson index > 2 (vs. ≤2)	--	--	--	--
*C. Intra-hospital mortality*				
- Co-infection vs. SARS-CoV-2 infection only	0.76 (0.24–2.41)	0.6	1.03 (0.28–3.81)	0.9
- Age, 1-year increase	--	--	1.07 (1.05–1.10)	<0.001
- Male gender	--	--	--	--
- Charlson index > 2 (vs. ≤2)	--	--	--	--
*D. ICU admission*	0.54 (0.41–5.74)	0.5	1.20 (0.24–5.95)	0.8
- Co-infection vs. SARS-CoV-2 infection only	--	--	1.1 (0.98–1.03)	0.9
- Age, 1-year increase	--	--	--	--
- Male gender	--	--	--	--
- Charlson index > 2 (vs. ≤2)				

OR: odds ratio; CI: confidence interval; ICU: intensive care unit.

**Table 3 microorganisms-10-01636-t003:** Selected characteristics and outcomes of the sample according to infection status: (a) SARS-CoV-2 infection only; and (b) SARS-CoV-2 with CP and/or MP co-infection.

	SARS-CoV-2 Infection Only	CP and/or MP Co-Infection	*p* **
Variables	(*n* = 194)	(*n* = 51)	
Male gender, %	51.0	52.9	0.8
Mean age (SD)	66.7 (18.1)	67.8 (17.0)	0.7
Median Charlson index at admission (IQR)	1.0 (3.0)	1.0 (2.0)	0.011
- % of patients with score ≥ 2	33.5	9.8	0.001
- % of patients with score ≥ 5	8.3	3.9	0.3
*Selected laboratory parameters **			
PLT, ×10^3^/µL	199 (87)	230 (125)	0.046
CRP, mg/dL	11.3 (44.5)	8.83 (7.37)	0.8
D-dimer, mg/L	1.96 (3.97)	2.98 (5.87)	0.8
Lactate, mmol/L	1.21 (0.72)	0.91 (0.36)	0.10
LDH, U/L	326 (146)	359 (137)	0.07
CPK, U/L	236 (617)	153 (166)	0.8
Procalcitonin, ng/mL	1.75 (5.48)	0.81 (1.71)	0.13
pO_2_, mmHg	76.8 (27.8)	64.5 (13.9)	0.02
pCO_2_, mmHg	33.5 (6.11)	31.9 (4.6)	0.2
*CT signs*			
Pneumonia, %	29.9	30.0	0.9
Median RSNA category (IQR)	3.0 (1.0)	3.0 (1.0)	0.9
- % patients with score = 3	64.4	81.4	0.15
Median CO-RADS category (IQR)	5.0 (2.0)	5.0 (1.0)	0.8
- % patients with score = 5	62.0	81.4	0.14
Median extent (IQR)	2.0 (2.0)	2.0 (2.0)	0.8
- % patients with score ≥ 3	31.3	44.2	0.11
Median consolidation (IQR)	1.0 (1.0)	1.0 (1.0)	0.9
- % patients with score ≥ 3	1.8	4.7	0.6
Median ground-glass opacity (IQR)	1.0 (1.0)	1.0 (2.0)	0.8
- % patients with score ≥ 3	6.8	11.6	0.3
Median crazy paving (IQR)	0.0 (1.0)	1.0 (1.0)	0.9
- % patients with score ≥ 3	6.8	9.3	0.6
Tree in bud ≥ 2, %	1.8	2.3	0.8
Pleural effusion (mono/bilateral), %	18.4	7.0	0.2
Pericardial effusion, %	1.2	0.0	--
Lymph node enlargement, %	17.2	20.9	0.6
*Outcomes*			
1-week mortality, %	7.3	3.9	0.4
30-days mortality, %	19.2	9.8	0.2
Intra-hospital mortality, %	25.9	17.7	0.2
Mean LOS (SD)	16.3 (18.5)	15.4 (14.2)	0.5
ICU admission, %	10.8	14.0	0.5

SD: standard deviation; IQR: interquartile range; PLT: platelets; CRP: C-reactive protein; pO_2_: partial pressure of oxygen; pCO_2_: partial pressure of carbon dioxide; CT: computed tomography; RSNA: Radiological Society of North America; CO-RADS: COVID-19 Reporting and Data System score; ICU: intensive care unit; LOS: length of stay. * All values are means (SD) when not otherwise reported. ** Chi-square test for categorical values; t-test and Kruskal–Wallis test for normally and non-normally distributed continuous variables, respectively.

**Table 4 microorganisms-10-01636-t004:** Univariate and multivariate analysis evaluating the potential predictors of each outcome in subjects with SARS-CoV-2 and both co-infections vs. SARS-CoV-2 infection alone.

*Outcomes*	Crude OR (95% CI)	*p*	Adjusted OR (95% CI)	*p*
*A. 1-week mortality*				
- Co-infection vs. SARS-CoV-2 infection only	0.52 (0.11–2.37)	0.4	1.12 (0.21–5.86)	0.9
- Age, 1-year increase	--	--	1.06 (1.01–1.11)	0.025
- Male gender	--	--	0.60 (0.19–1.91)	0.4
- Charlson index > 2 (vs. ≤2)	--	--	2.04 (0.62–6.75)	0.2
*B. 30-day mortality*				
- Co-infection vs. SARS-CoV-2 infection only	0.46 (0.17–1.23)	0.12	0.65 (0.20–2.13)	0.5
- Age, 1-year increase	--	--	1.05 (1.02–1.08)	0.002
- Male gender	--	--	1.16 (0.55–2.44)	0.7
- Charlson index > 2 (vs. ≤2)	--	--	1.95 (0.88–4.32)	0.10
*C. Intra-hospital mortality*				
- Co-infection vs. SARS-CoV-2 infection only	0.61 (0.28–1.34)	0.2	1.11 (0.42–2.90)	0.8
- Age, 1-year increase	--	--	1.06 (1.03–1.08)	<0.001
- Male gender	--	--	1.03 (0.52–2.05)	0.9
- Charlson index > 2 (vs. ≤2)	--	--	2.16 (1.04–4.45)	0.04
*D. ICU admission*				
- Co-infection vs. SARS-CoV-2 infection only	1.34 (0.54–3.36)	0.5	0.85 (0.26–2.80)	0.8
- Age, 1-year increase	--	--	1.01 (0.99–1.04)	0.3
- Male gender	--	--	3.22 (1.20–8.59)	0.020
- Charlson index >2 (vs. ≤2)	--	--	0.24 (0.06–0.94)	0.04

OR: odds ratio; CI: confidence interval; ICU: intensive care unit.

## Data Availability

The datasets generated and/or analysed during the current study are not publicly available due to privacy policy but are available from the corresponding author on reasonable request.

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
