# Peer review of "Role of Intracellular Pulmonary Pathogens during SARS-CoV-2 Infection in the First Pandemic Wave of COVID-19: Clinical and Prognostic Significance in a Case Series of 1200 Patients"

_microorganisms, 2022, doi:10.3390/microorganisms10081636_

Round 1

Reviewer 1 Report

This study conducted by Drs. Guarino and colleagues reported the clinically important issues regarding the complicated co-infectious manifestations of SARS-CoV-2.

This is a well-written manuscript with a presentation/discussion of SARS-CoV-2 infected patients with or without airway bacterial infection especially focusing on the pathogens related to atypical community acquired pneumonia.

According to the authors' results and conclusions, the reviewer summarized this study's findings as follows: 1) the prevalence of either MP or CP co-infection was about 21% in hospitalized patients with SARS-CoV-2 during the first pandemic wave. 2) CT finding on admission showed that "Pneumonia" existed at a similar rate of 30% in both groups (SARS-CoV-2 w/wo co-infection). 3) the recovered time (length of hospital stay) and mortality rate did not differ between groups, probably due to proper administration of azithromycin from the day of admission in all hospitalized patients).

The general ideas are promising, and the purpose/methods are clearly mentioned; however, the reviewer raises concerns and would like to hear the authors' comments.

1) Please mention the relatively low rate of pneumonia in this study group.

Clinicians have experienced much more (> 50%) pneumonia in patients with SARS-CoV-2 during the first pandemic wave even within three days from symptom onset (Morikawa M et al. Intern Med 2020, DOI:10.2169/internalmedicine.5528-20). So, the reviewer assumed that the patients enrolled in this study would vias toward mild COVID-19, which should be mentioned in the Discussion.

2) Please clarify the prevalence of each atypical intracellular pathogen in 1204 patients.

Of the 245 patients with SARS-CoV-2, 32 (12.6%) had co-infected with MP, and 19 (7.8%) had co-infected with CP. Although the reviewer got the impression a bit higher existence of CP, this would be reasonable as a community acquired pneumonia reported by Marchello et al. (Ann Fam Med 2016;14:552, DOI: 10.1370/afm.1993). In this regard, the reviewer assumes that the readers would like to know the rate of existence of each bacteria in the remaining 959 patients. If the results in the remaining patients were similar to those observed in COVID-19 patients, we could speculate the event of SARS-CoV-2 is a secondary infection, which would be an extremely important finding in clinical settings.

3) Of the patients having pneumonia (58 patients in SARS-CoV-2 only and 15.3? patients in SARS-CoV-2 co-infected with either MP or CP), were there any differences in clinical manifestations and outcomes observed?

The reviewer recommends considering a comparison of clinical manifestations and outcomes in moderate (to severe) COVID-19 patients (having pneumonia) with or without co-infection (even if the authors would be worried that the number of involved patients was small). Furthermore, it would attract strong interest in clinicians taking care of moderate to severe COVID-19 patients routinely administering antibiotics of macrolides (azithromycin) or respiratory quinolone since admission. So, if a poorer outcome in patients with SARS-CoV-2 plus atypical bacterial infection is expected, especially when developed in pneumonia, the clinician should consider earlier administration of azithromycin in the outpatient clinic.

4) More than two groups' comparisons should be analyzed by using ANOVA for parametric distribution.

              In Table 1, the legend describes that the statistical method comparing the three groups was t-test for normally distributed continuous variables. It should be clarified.

Minor concern:

P1. L27. Please consider spelling out C. pneumoniae (Chlamydia pneumoniae) and M. pneumoniae (Mycoplasma pneumoniae) as these appear first in the abstract.

P2. L81. Please refer/show the result of IL-6 if the authors mentioned it in Materials and Methods section.

P2. L89. Please remove the hyphen.

P3. L97. Please remove the hyphen as well.

P7. Table 3. The description of "CP+MP co-infection" might confuse.

Reviewer 2 Report

The manuscript “Role of intracellular pulmonary pathogens during SARS-CoV-2  

infection in the first pandemic wave of COVID-19: clinical and prognostic significance in a case series of 1200 patients” from Matteo Guarino et al. studies the the existence of bacterial co-infections and their possible role as cofactors worsening COVID-19-related clinical manifestations, however, there are several concerns on this manuscript:

1. The authors describe the role of bacterial co-infection in patients with COVID-19, how authors distinguish bacterial co-infection and secondary infection during the course of the illness? 

2. As author summarized in limitations, the patients sample are less even in group B and group C, and the methods in this article to divide patients into three subgroups: group A (SARS-CoV-2 infection only); group B (SARS-CoV-2 and MP); and group C (SARS-CoV-2 and CP) based on using a serological assay for the determination of IgG and IgM class antibodies. First, some patients do not have sufficient levels of M. pneumoniae IgM antibodies to allow for detection during the early stages of acute infection or reinfection, and some patients may have detectable levels of M. pneumoniae IgM for several months after M. pneumoniae infection, moreover, since many of the pandemic viral pneumonias have similar clinical and radiological features that may make it difficult to distinguish from other common bacterial (such as pneumococcal, staphylococcal and Klebsiella spp.), viral (seasonal respiratory viruses), or fungal (e.g. Pneumocystis jirovecii) causes of pneumonia. So are there any other methods used in regard to dividing to different group?

3. Except radiological pulmonary infiltrates, C-reactive protein also often used to differentiate bacterial from viral causes in community-acquired pneumonia. Did author detect antimicrobial-associated C-reactive protein? How it looks among 3 groups? 

4. No IL-6 test in results, but Materials and Methods mentioned as Line 81

5. Introduction part is a little bit short, please give more information.

Reviewer 3 Report

The authors are dealing with the effect of intracellular pathogens on Covid- 19 patients.

The paper is generally well written but several things are missing and others need amelioration

Round 2

Reviewer 1 Report

The authors have considered all the questions, comments and suggestions, and thus the revised manuscript has been substantially improved.

Author Response

We wish to thank the Reviewer for the insightful comments, which indeed increased the quality of our work

Reviewer 2 Report

The questions has been well addressed.

Author Response

(The authors gave the same response as above.)

Reviewer 3 Report

A correction in the abstract, Furthermore you ignored the point that in the discussion you should start with a recap of you results.

Author Response

We kindly apologize for the mistake. We added a short sentence as “recap” of the main results of our study. This sentence has been introduced at the beginning of the Discussion.